# Smoothed Analysis of Discrete Tensor Decomposition and Assemblies of Neurons

**Nima Anari**
Computer Science
Stanford University
anari@cs.stanford.edu

**Constantinos Daskalakis**
EECS
MIT
costis@csail.mit.edu

**Wolfgang Maass**
Theoretical Computer Science
Graz University of Technology
maass@igi.tugraz.at

**Christos H. Papadimitriou**
Computer Science
Columbia University
christos@cs.columbia.edu

**Amin Saberi**
MS&E
Stanford University
saberi@stanford.edu

**Santosh Vempala**
Computer Science
Georgia Tech
vempala@gatech.edu

## Abstract

We analyze linear independence of rank one tensors produced by tensor powers of randomly perturbed vectors. This enables efficient decomposition of sums of high-order tensors. Our analysis builds upon Bhaskara et al. [3] but allows for a wider range of perturbation models, including discrete ones. We give an application to recovering assemblies of neurons.

Assemblies are large sets of neurons representing specific memories or concepts. The size of the intersection of two assemblies has been shown in experiments to represent the extent to which these memories co-occur or these concepts are related; the phenomenon is called *association of assemblies*. This suggests that an animal's memory is a complex web of associations, and poses the problem of recovering this representation from cognitive data. Motivated by this problem, we study the following more general question: Can we reconstruct the Venn diagram of a family of sets, given the sizes of their $\ell$-wise intersections? We show that as long as the family of sets is randomly perturbed, it is enough for the number of measurements to be polynomially larger than the number of nonempty regions of the Venn diagram to fully reconstruct the diagram.

## 1 Introduction

Tensor decomposition is one of the key algorithmic tools for learning many latent variable models [1, 5, 14, 19]. In practice, tensor decomposition methods based on gradient descent and power method have been observed to work well [9, 16]. Theoretically, determining the minimum number of rank one components in the tensor decomposition is known to be NP-hard in the worst case [11, 12], so usually tensor decomposition is analyzed in the average case. Several algorithms have been analyzed in the average case, where the input tensor is produced according to some probabilistic model, for example see Bhaskara et al. [3], De Lathauwer et al. [7], Goyal et al. [10] as well as sum-of-squares-based algorithms like Barak et al. [2], Ge and Ma [8], Hopkins et al. [13], Ma et al. [18].

The average case models studied in the literature generally fall into two categories. They either assume components of the tensor are fully random, i.e., generated from a known distribution (e.g., Gaussian), or they follow a smoothed analysis setting where some adversarially chosen instance is perturbed by random noise, see for example Bhaskara et al. [3], Goyal et al. [10], Ma et al. [18]. Our work falls into the second category.

We build upon the framework used in Bhaskara et al. [3] which reduces decomposing sums of rank one tensors to showing robust linear independence of related rank one tensors, by using Jennrich's algorithm, also known as Chang's lemma [5, 17]. The main departing point of our work is our smoothed analysis of linear independence, which we base on a new notion we call echelon trees, a generalization of Gaussian elimination and echelon form to high-order tensors, which might be of independent interest. We also get improved guarantees compared to Bhaskara et al. [3] when the tensors are of high enough order.

The main feature of our analysis is that it can handle discrete perturbations. To illustrate, suppose that vectors $X_1, \ldots, X_m \in \mathbb{R}^n$ are drawn from some unknown distribution and our goal is to recover them by (noisily) observing $\sum_i X_i^{\otimes \ell}$ for small values of $\ell$. Bhaskara et al. [3] showed that up to constant factor blow-ups in $\ell$ an efficient algorithm can do this as long as $X_i^{\otimes \ell}$ are linearly independent in a robust sense. Note that the set of vector tuples $(X_1, \ldots, X_m)$ for which $X_1^{\otimes \ell}, \ldots, X_m^{\otimes \ell}$ are linearly dependent can be defined by polynomial equations, using determinants, and is therefore an algebraic variety. As long as $m \ll n^{\ell}$, this variety will have dimension smaller than the whole space, so we expect most vector tuples to fall outside. Bhaskara et al. [3] showed that starting from an arbitrary set of vectors $X_1, \ldots, X_m$, by adding Gaussian noise, the new tuple will lie far away from this variety. Our analysis on the other hand, handles a much wider class of perturbations. For example, if each $X_i$ is independently chosen at random from a "large enough" discrete set such as the vertices of an arbitrary hypercube, we show that with very high probability the resulting tensors are linearly independent, again in a robust sense.

For our main application, described in the next section, it is important to assume components of the tensor come from a discrete set.

## 1.1 Assemblies of neurons and recovering sparse Venn diagrams

Experiments by neuroscientists over the past three decades [21] have identified neurons which are selectively activated when a real-world object[1] is seen (or more generally sensed). It is now widely accepted [4] that these neurons are part of large *cell assemblies*, stable sets of highly interconnected neurons whose firing (more or less simultaneous and in unison) is tantamount to a cognitive event such as the sensing or imagining of a person, or of a word or concept (hence the other common name "concept cells").

In a recent experiment [15], a neuron firing when one real-world entity is seen (say, the Eiffel tower) but not another (e.g., Barak Obama) may start firing on presentation of an image of Obama *after a visual experience associating the two* — for example, a picture of Obama in front of the Eiffel tower. This experiment has taught us that assemblies seem to be "mobile" and able to intersect in complex ways reflecting perceived varying degrees of associations between the corresponding entities. The stronger the association between the entities, the larger the intersection will be of the corresponding assemblies. During one's life, presumably a complex mesh of entities and associations will be created, of some degree of permanence, reflecting the sum total of one's cognitive experiences.

All said, this complex mesh of memories in somebody's brain can be modeled as a Venn diagram where each set or assembly consists of neurons firing for a particular concept, and each region of the Venn diagram, a minimal set obtained from an intersection of assemblies and their complements, represents a class of neurons behaving the same way towards all concepts.

Alternatively to the Venn diagram, one may record associations between assemblies in a hypergraph. The entities are the sets or nodes, and the edges reflect associations between the nodes. Furthermore, the hypergraph representing a person's state of knowledge can be adorned *with edge weights* reflecting the degree of affinity between a set of nodes (or equivalently, the size of the intersection of their corresponding sets).

This gives rise to several natural questions. The first question concerns reconstruction. How many experiments or observations are needed to identify the structure of cell assembly intersections, or in other words the Venn diagram? Here, we make two crucial assumptions. First, we assume that we can only measure the degree of association between a small number of entities or concepts. Second, the total number of classes of neurons (which behave similarly in response to stimuli) is bounded. In the language of sets, we assume the number of non-empty regions of the Venn diagram is upper

bounded by some number $m$ and we can measure the sizes of $k$-wise intersections of any $k$ of our $n$ sets for $1 \leq k \leq \ell$ for some small $\ell$. We also allow for measurement errors.

Our main result here is as follows: As long as the cell assemblies are slightly randomly perturbed, and as long as the number of measurements, $\binom{n}{1} + \binom{n}{2} + \cdots + \binom{n}{\ell}$, is polynomially larger than the number of nonempty regions of the Venn diagram, $m$, we can fully reconstruct the Venn diagram. The perturbation of cell assemblies, a process which likely occurs naturally in the brain, is a mild assumption that we need in order to escape idiosyncratic cases. We solve the problem of reconstructing the Venn diagram by casting it as a tensor decomposition problem where the elements of the decomposition come from high order tensors of the vertices of the hypercube.

We also explore a simpler graph-theoretic model of assembly association, motivated by more recent experimental findings [6, 15]: Assume that all assemblies have the same size $K$, and that two assemblies are associated if their intersection is of size at least $b$, and are not associated if the intersection is less than another threshold $a < b$; the results of De Falco et al. [6], Ison et al. [15] suggest that $a$ is 4% of $K$, while $b$ is 8% of $K$. We show that an unreasonably rich and complex family of graphs can be realized by associations (roughly, any graph of degree $O(K/a)$).

## 1.2 Problem formulation

Suppose that we have a Venn diagram formed by some $n$ sets $\mathcal{S}_1, \ldots, \mathcal{S}_n$. We will assume that this Venn diagram has at most $m$ nonempty regions. For our main application, each set $\mathcal{S}_i$ corresponds to neurons that respond to a particular stimuli, so we are assuming that there are at most $m$ classes of neurons. We let $\mathcal{U}$ denote the set of neuron classes. We also have a weight function $w : \mathcal{U} \to \mathbb{R}_{\geq 0}$ representing the sizes of various classes. Each set $\mathcal{S}_i \subseteq \mathcal{U}$ is an assembly and $w(\mathcal{S}_i) = \sum_{u \in \mathcal{S}_i} w(u)$ is its weight. Our main question is the following:

**Question 1.** *Given the sizes of $\ell$-wise intersections of $\mathcal{S}_1, \ldots, \mathcal{S}_n$ for some constant $\ell$, i.e., $w(\mathcal{S}_{i_1} \cap \cdots \cap \mathcal{S}_{i_\ell})$ for all $i_1, \ldots, i_\ell \in [n]$, can we recover the full Venn diagram of $\mathcal{S}_1, \ldots, \mathcal{S}_n$, i.e., the weight of all intersections formed by these sets and their complements?*

Our main result is that as long as the set memberships of elements are slightly perturbed to avoid worst case scenarios, and as long as $n^\ell$ is polynomially larger than $m = |\mathcal{U}|$, the answer is yes and moreover there is an efficient algorithm that performs recovery. Our algorithm is also robust to inverse polynomial noise in the input.

We pose the question as a tensor decomposition problem in the following way: To each element $u \in \mathcal{U}$ assign a vector $\chi(u) \in \{0, 1\}^n$, where $\chi(u)_i$ indicates whether $u \in \mathcal{S}_i$. Then the entries of the following tensor capture all $\ell$-wise intersections:

$$T = \sum_{u \in \mathcal{U}} w(u) \underbrace{\chi(u) \otimes \cdots \otimes \chi(u)}_{\ell \text{ times}}.$$

For simplicity of exposition, we assume weights are all equal to 1, but our results easily generalize, since each weight $w(u)$ can be absorbed into $\chi(u)^{\otimes \ell}$.

## 2 Notations and preliminaries

We denote the set $\{1, \ldots, n\}$ by $[n]$. For a matrix $A$, we denote the minimum and maximum singular values of $A$ by $\sigma_{\min}(A)$ and $\sigma_{\max}(A)$. We use $\langle \cdot, \cdot \rangle$ to denote the standard inner product.

We denote the tensor product of two vectors $\chi \in \mathbb{R}^n$ and $\chi' \in \mathbb{R}^m$ by $\chi \otimes \chi'$ which belongs to $\mathbb{R}^n \otimes \mathbb{R}^m \simeq \mathbb{R}^{n \times m}$. We use the notation $\chi^{\otimes \ell}$ to denote

$$\underbrace{\chi \otimes \cdots \otimes \chi}_{\ell \text{ times}}.$$

By abuse of notation we identify tensors $T \in \mathbb{R}^{n_1} \otimes \cdots \otimes \mathbb{R}^{n_\ell}$ with multilinear maps from $\mathbb{R}^{n_1} \times \cdots \times \mathbb{R}^{n_\ell}$ to $\mathbb{R}$. In other words we let $T(v_1, \ldots, v_\ell)$ denote $\langle T, v_1 \otimes \cdots \otimes v_\ell \rangle$. We also use the notation $T(\cdot, v_2, \ldots, v_\ell)$ to denote the multilinear map from $\mathbb{R}^{n_1}$ to $\mathbb{R}$ given by:

$$T(\cdot, v_2, \ldots, v_\ell)(v_1) = T(v_1, \ldots, v_\ell).$$

In general we can use $\cdot$ in place of any of the arguments of $T$. So for example $T(\cdot, \cdot, v_3, \ldots, v_\ell)$ is interpreted as living in $\mathbb{R}^{n_1} \otimes \mathbb{R}^{n_2}$. With a slight abuse of notation we let some of the inputs of $T$ be merged together by tensor operations. In other words we let $T(v_1 \otimes v_2, v_3, \ldots, v_\ell)$ be the same as $T(v_1, \ldots, v_\ell)$.

We use $e_1, \ldots, e_n$ to denote the standard basis of $\mathbb{R}^n$. For a tuple of coordinates $I = (i_1, \ldots, i_\ell)$ we let $e_I$ denote $e_{i_1} \otimes \cdots \otimes e_{i_\ell}$. With this notation, the entry corresponding to coordinate $(i_1, \ldots, i_\ell)$ of a tensor $T$ can be written as $T(e_I) = T(e_{i_1}, \ldots, e_{i_\ell})$.

## 3  Tensor decomposition

Suppose that we have a finite universe $\mathcal{U}$ of elements with a vector $\chi(u) \in \mathbb{R}^n$ assigned to each $u \in \mathcal{U}$. Our goal is to recover $\chi(u)$'s by observing $\sum_u \chi(u)^{\otimes \ell}$. A necessary condition is for $\chi(u)^{\otimes \ell}$'s to be linearly independent, otherwise it is an easy exercise to show that there is another decomposition $\sum_u (c_u \chi(u))^{\otimes \ell}$ for some positive weights $\{c_u\}_{u \in \mathcal{U}}$ not all equal to 1. The framework introduced by Bhaskara et al. [3] shows that linear independence is not just necessary, but up to a constant factor blow-up in $\ell$, it is sufficient. A more detailed account is given in supplementary materials.

We also use another trick from this framework which allows us to replace symmetric tensors $\chi(u)^{\otimes \ell}$ with asymmetric ones. If we divide the coordinates $[n]$ into $\ell$ roughly-equal sized parts $I_1, \ldots, I_\ell$ and define $\chi(u)^{(i)}$ to be the projection of $\chi(u)$ onto the $i$-th part, then $\chi(u)^{(1)} \otimes \cdots \otimes \chi(u)^\ell$ is a subtensor of $\chi(u)^{\otimes \ell}$. So linear independence of these tensors proves linear independence of $\chi(u)^{\otimes \ell}$'s. The advantage of this trick is that when we introduce perturbations to $\chi(u)^{(1)}, \ldots, \chi(u)^{(\ell)}$, we do not have to worry about consistently perturbing the same coordinates and we can potentially use independent randomness. For simplicity of notation, from here on, we use $n$ (as opposed to $n/\ell$) to denote the dimension of each $\chi(u)^{(i)}$. So now we can work with the following tensor:

$$T = \sum_{u \in \mathcal{U}} \chi(u)^{(1)} \otimes \cdots \otimes \chi(u)^{(\ell)}.$$

Our main result is that the components of this sum are robustly linearly independent, assuming the components $\chi(u)^{(i)}$ are randomly perturbed. We remark that this implies robust linear independence of $\{\chi(u)^{\otimes \ell}\}_{u \in \mathcal{U}}$ as well, so we can recover them from the sum $\sum_{u \in \mathcal{U}} \chi(u)^{\otimes \ell}$.

We first define our model of perturbations:

**Definition 2.** Assume that a vector $X \in \mathbb{R}^d$ is drawn according to some distribution $\mathcal{D}$. We call $\mathcal{D}$ a $(\delta, p)$-nondeterministic distribution if for every coordinate $i \in [d]$ and any interval of the form $(t - \delta, t + \delta)$ we have

$$\mathbb{P}[X_i \in (t - \delta, t + \delta) \mid X_{-i}] \leq p,$$

where $X_{-i}$ represents the projection of $X$ onto the coordinates $[d] - \{i\}$.

For a set of random vectors $\{X_i\}$, we call their joint distribution $(\delta, p)$-nondeterministic iff their concatenation is $(\delta, p)$-nondeterministic. In our setting, we will assume that for each $u \in \mathcal{U}$, the vectors $\chi(u)^{(1)}, \ldots, \chi(u)^{(\ell)}$ are chosen from a $(\delta, p)$-nondeterministic distribution.

Two examples of $(\delta, p)$-nondeterministic perturbations can be obtained as follows:

*Example* 3. Suppose that each $\chi(u)^{(i)}$ is chosen adversarially from $\{0, 1\}^n$, but then each bit is independently flipped with some probability $q$. This distribution is $(\frac{1}{2}, \max(q, 1 - q))$-nondeterministic.

*Example* 4. Suppose that each $\chi(u)^{(i)}$ is chosen adversarially from $\mathbb{R}^n$, but a standard Gaussian noise of total variance $\rho^2$ is added to each one. Then for any $\delta > 0$, this distribution is $(\delta, \mathrm{erf}(\sqrt{n}\delta/\rho))$-nondeterministic.

Gaussian perturbations are the model used in Bhaskara et al. [3]. Our main result is the following:

**Theorem 5.** *Assume that for each* $u \in \mathcal{U}$*, the concatenation of the $n$-dimensional vectors* $\{\chi(u)^{(i)}\}_{i \in [\ell]}$ *is drawn from a distribution $\mathcal{D}$ that is $(\delta, p)$-nondeterministic. Let $A$ be the matrix whose columns are given by flattened $a(u) = \chi(u)^{(1)} \otimes \cdots \otimes \chi(u)^\ell$ for various $u$. Then, assuming $|\mathcal{U}| \leq (cn)^\ell$, we have*

$$\mathbb{P}[\sigma_{\min}(A) < (\delta/n)^\ell] \leq n^{2\ell} p^{(1-c)n}.$$

This theorem shows how the $(\delta, p)$-nondeterministic property ensures robust linear independence. To prove it, we use a strategy similar to Bhaskara et al. [3], by proving a bound on the leave-one-out distance. The leave-one-out distance is closely related to $\sigma_{\min}(A)$, and only differs from it by a factor of at most $\sqrt{|\mathcal{U}|} \leq n^{\ell/2}$ [3]. It is enough to prove that for any fixed $u$

$$\text{dist}\left(a(u), \text{span}\{a(u')\}_{u' \in \mathcal{U}-\{u\}}\right) \geq (\delta/\sqrt{n})^\ell$$

with probability at least $1 - n^\ell p^{(1-c)n}$. Here dist measures the distance of a vector to the closet point in a linear subspace. A union bound implies the leave-one-out distance for all $u$ is large. As in Bhaskara et al. [3], we simplify the analysis by treating $\text{span}\{a(u')\}_{u' \in \mathcal{U}-\{u\}}$ as a generic linear subspace $V \subseteq (\mathbb{R}^n)^{\otimes \ell}$, and only using the fact that $\dim(V) < (cn)^\ell$. Noting that $n^\ell \geq 1 + n + n^2 + \cdots + n^{\ell-1}$, it is enough to prove the following

**Lemma 6.** *Assume that vectors $\chi^{(1)}, \ldots, \chi^{(\ell)}$ are drawn according to a $(\delta, p)$-nondeterministic distribution. Further assume that $V \subseteq (\mathbb{R}^n)^{\otimes \ell}$ is a subspace of dimension at most $(cn)^\ell$. Then*

$$\mathbb{P}\left[\text{dist}\left(\chi^{(1)} \otimes \cdots \otimes \chi^{(\ell)}, V\right) < (\delta/\sqrt{n})^\ell\right] \leq (1 + n + n^2 + \cdots + n^{\ell-1})p^{(1-c)n}.$$

In the rest of this section we prove lemma 6.

Let $W = V^\perp \subseteq (\mathbb{R}^n)^{\otimes \ell}$ be the linear subspace of all tensors that vanish on $V$, or in other words have zero dot product with every member of $V$. Then $\dim(W) \geq (1 - c^\ell)n^\ell$. We will show that with high probability there is an element $T \in W$ such that $\|T\| \leq n^{\ell/2}$ and

$$\langle T, \chi^{(1)} \otimes \cdots \otimes \chi^{(\ell)} \rangle = T(\chi^{(1)}, \ldots, \chi^{(\ell)}) \geq \delta^\ell.$$

This implies that

$$\text{dist}\left(\chi^{(1)} \otimes \cdots \otimes \chi^{(\ell)}, V\right) \geq \frac{\delta^\ell}{\|T\|} = n^{-\ell/2}\delta^\ell = (\delta/\sqrt{n})^\ell,$$

and the proof would be complete.

We find it instructive to first prove this fact for $\ell = 1$ and then for general $\ell$.

## 3.1  The case $\ell = 1$

*Proof of lemma 6 for $\ell = 1$.* We will generate a sequence $T_1, \ldots, T_{\dim(W)} \in W$, such that $\|T_i\|_\infty \leq 1$ for all $i$. This ensures that $\|T_i\| \leq \sqrt{n}$. We will then show that

$$\mathbb{P}[\exists i : |T_i(\chi)| \geq \delta] \geq 1 - p^{(1-c)n}. \tag{1}$$

We will first pick $T_1$ to be any nonzero element of $W$. By rescaling, we can assume that $\|T_1\|_\infty = 1$ and that $T_1(e_j) = 1$ for some $j$. Let us call $j$ the pivot point of $T_1$. By rearranging the coordinates we can assume without loss of generality that $j = 1$. In other words $T_1(e_1) = 1$ and $\|T_1\|_\infty = 1$.

In order to pick $T_2$, consider the subspace $\{T \in W \mid T(e_1) = 0\}$. This subspace has dimension at least $\dim(W) - 1$, and we can pick $T_2$ to be any nonzero element of it. As before, we can without loss of generality and by scaling assume that $T_2(e_2) = 1$ and $\|T_2\|_\infty = 1$.

When picking $T_i$, we pick any nonzero element of $\{T \in W \mid T(e_j) = 0 \ \forall j < i\}$ and by rescaling and rearranging the coordinates assume that $T_i(e_i) = 1$ and $\|T_i\|_\infty = 1$. Thus we make sure that the pivot point of $T_i$ is $i$. A keen observer would notice that $T_1, \ldots, T_{\dim(W)}$ can also be obtained by a modified Gaussian elimination procedure run on some basis of the space $W$.

Now that we have fixed $T_1, \ldots, T_{\dim(W)}$ it remains to prove eq. (1).

To do this, let us fix the coordinates of the random vector $\chi = \chi^{(1)}$ one-by-one, starting from $\chi_n$ and going backwards to $\chi_1$. Once we have fixed $\chi_{\dim(W)+1}, \ldots, \chi_n$ we can argue about the probability of the event $|T_{\dim(W)}(\chi)| < \delta$. Since $T_{\dim(W)}(e_i) = 0$ for $i < \dim(W)$, we have

$$T_{\dim(W)}(\chi) = \chi_{\dim(W)} + T_{\dim(W)}(e_{\dim(W)+1})\chi_{\dim(W)+1} + \cdots + T_{\dim(W)}(e_n)\chi_n.$$

But $t := T_{\dim(W)}(e_{\dim(W)+1})\chi_{\dim(W)+1} + \cdots + T_{\dim(W)}(e_n)\chi_n$ is a constant once we have fixed $\chi_{\dim(W)+1}, \ldots, \chi_n$. So $|T_{\dim(W)}(\chi)| < \delta$ if and only if $\chi_{\dim(W)} \in (-t - \delta, -t + \delta)$. Because $\chi$ is

distributed according to a $(\delta, p)$-nondeterministic distribution, this event happens with probability at most $p$. In other words

$$\mathbb{P}[|T_{\dim(W)}(\chi)| < \delta] \leq p.$$

If this event does not occur, we are already done. Otherwise we can condition on $\chi_{\dim(W)}, \ldots, \chi_n$, and look at the event $|T_{\dim(W)-1}(\chi)| < \delta$. Once we condition on $\chi_{\dim(W)}$, this event becomes independent of the previous event and we can again upperbound its probability by $p$. So we have

$$\mathbb{P}[|T_{\dim(W)-1}(\chi)| < \delta \mid |T_{\dim(W)}(\chi)| < \delta] \leq p$$

which implies

$$\mathbb{P}[|T_{\dim(W)-1}(\chi)| < \delta \wedge |T_{\dim(W)}(\chi)| < \delta] \leq p^2.$$

By continuing this, in the end we get

$$\mathbb{P}[\wedge_{i=1}^{\dim(W)} |T_i(\chi)| < \delta] \leq p^{\dim(W)} \leq p^{(1-c)n},$$

which is the complement of eq. (1).                                                                        $\square$

## 3.2 The general case

Here we describe a structure that we name *echelon tree*. This definition is motivated by the Gaussian elimination procedure for matrices that produces an *echelon form*. Our definition can be seen as a generalization of this form for tensor spaces.

We first describe an *index tree* for $\mathbb{R}^{n_1 \times \cdots \times n_\ell}$: Consider an abstract rooted tree $\mathcal{T}$ of height $\ell$ where the nodes at level $k$ are labeled by *different* partial indices from $[n_1] \times [n_2] \times \cdots \times [n_k]$; the root has the empty label and resides at level $0$, and all leaves reside at level $\ell$. We require the indices to be consistent with the tree structure, i.e., all children (and by extension descendants) of a node labeled $I = (i_1, \ldots, i_k)$ must contain $I$ as the prefix of their label. We further assume that $\mathcal{T}$ is ordered, i.e., each node of $\mathcal{T}$ has an ordering over its children. This enables us to talk about post-order traversal of the tree, a linear ordering of the nodes of the tree, which we denote by the binary relation $\prec$. For two nodes labeled $I$ and $J$, we let $I \prec J$ exactly when (i) $I$ is a descendant of $J$ or (ii) there are ancestors $I', J'$ of $I, J$ with a common parent who places $I'$ before $J'$ (according to the ordering induced by the parent on its children).

**Definition 7.** An index tree for $\mathbb{R}^{n_1 \times \cdots \times n_\ell}$ is a height $\ell$ tree $\mathcal{T}$ of partial indices together with a post-traversal ordering $\prec$ on its nodes as described above.

We emphasize that nodes of an index tree have different labels, so we consider the partial indices the same as the nodes. For example, an index tree of height 1 is identical to an ordered list $i^{(1)}, \ldots, i^{(s)}$ of elements in $[n_1]$, with no repetitions allowed. Next we define an echelon tree.

**Definition 8.** An echelon tree is an index tree where each leaf $I$ is additionally labeled by an element $T_I \in \mathbb{R}^{n_1 \times \cdots \times n_\ell}$. We require that $T_I(e_I) \neq 0$ and that for every node $J$ that appears before $I$ in the post-order traversal, i.e., $J \prec I$, the following identity to hold:

$$T_I(e_J, \cdot, \ldots, \cdot) = 0.$$

Note that the identity in the above definition is requiring an entire sub-array of $T_I$ to be zero. For example a height 1 echelon tree is a list of unique indices $i^{(1)}, \ldots, i^{(s)}$ of $[n_1]$ together with vectors $T^{(1)}, \ldots, T^{(s)} \in \mathbb{R}^{n_1}$ such that $T^{(j)}$ has zeros in the $i^{(1)}, \ldots, i^{(j-1)}$ entries and has a nonzero $i^{(j)}$-th entry. Notice the similarity to the echelon form obtained by Gaussian elimination in a matrix. In particular, for a height 1 echelon tree, the vectors $T^{(1)}, \ldots, T^{(s)}$ must be linearly independent.

We say that $\mathcal{T}$ is an echelon tree for the linear subspace $W \subseteq \mathbb{R}^{n_1 \times \cdots \times n_\ell}$ if for all leaves $I$, we have $T_I \in W$. Notice that we can collapse or flatten consecutive levels of an echelon tree, and the result would remain an echelon tree. In this operation, nodes of a particular level $i$ are removed, and each orphaned node of level $i + 1$ is assigned to its grandparent (of level $i - 1$). We then treat the indices as coming from $[n_1] \times \cdots \times [n_i n_{i+1}] \times \cdots \times [n_\ell]$, i.e., we merge the $i, i + 1$-st level indices. This also corresponds to partially flattening tensors $T_I$ and considering them as elements of $\mathbb{R}^{n_1 \times \cdots \times n_i n_{i+1} \times \cdots n_\ell}$. It is easy to check that these operations preserve the properties in definition 8:

**Fact 9.** *Collapsing an echelon tree at level $i$ produces an echelon tree.*

The main question we would like to address here is how large of an echelon tree can be constructed for a subspace $W$. For example, for $\mathbb{R}^{n_1 \times \cdots \times n_\ell}$ one can get a *full* tree, where nodes at level $i-1$ have branching factor $n_i$, by simply placing the standard basis for $\mathbb{R}^{n_1 \times \cdots \times n_\ell}$ at the leaves. We measure the size of a tree by its *fractional branching* factor.

**Definition 10.** An echelon tree $\mathcal{T}$ has fractional branching $(\alpha_1, \ldots, \alpha_\ell) \in [0,1]^\ell$ if each node $I$ at level $i-1$ has at least $\alpha_i n_i$ children. For a single number $\alpha \in [0,1]$, we say $\mathcal{T}$ has fractional branching $\alpha$ when it has fractional branching $(\alpha, \alpha, \ldots, \alpha)$.

Note that fractional branching $\alpha$ implies that the tree has at least $\alpha^\ell n_1 \ldots n_\ell$ leaves. On the other hand, repeated applications of fact 9 on the echelon tree would produce a height 1 echelon tree, and we have already observed that the vectors assigned to the leaves in such a tree must be linearly independent. So this implies that $\dim(W) \geq \alpha^\ell n_1 \ldots n_\ell$. There is a partial inverse to this statement: If $W \subseteq \mathbb{R}^{n_1 \times \cdots \times n_\ell}$ has dimension $(1 - c^\ell) \cdot n_1 \ldots n_\ell$, then there is an echelon tree with fractional branching $1 - c$ for $W$. However, this fact is not "robust", since the elements of $W$ assigned to the leaves can have arbitrarily small or large entries. Instead we prove the following:

**Theorem 11.** *If $W \subseteq \mathbb{R}^{n_1 \times \cdots \times n_\ell}$ has dimension $(1 - c^\ell) \cdot n_1 \ldots n_\ell$, then there is an echelon tree with fractional branching $1 - c$ for $W$ such that for every leaf $I$ we have $\|T_I\|_\infty = 1$ and $|T_I(e_I)| = 1$.*

Let us see first see why theorem 11 is enough to prove lemma 6.

*Proof of lemma 6 for general $\ell$.* Note that $\|T_I\|_\infty = 1$ implies that $\|T_I\| \leq n^{\ell/2}$. So it suffices to show that $T_I(\chi^{(1)}, \ldots, \chi^{(\ell)}) \geq \delta^\ell$ for some $I$ with high probability.

Let us say that an echelon tree is $x$-large when $|T_I(e_I)| \geq x$ for all leaves $I$. Theorem 11 guarantees that the echelon tree produced by it is 1-large.

Our strategy is to fix $\chi^{(\ell)}, \chi^{(\ell-1)}, \ldots, \chi^{(1)}$ in that order, and simultaneously reduce the height of our echelon tree by 1 each time. When we fix $\chi^{(\ell)}$, we can get a smaller echelon tree in the following way: For each leaf $I$ in the echelon tree, consider the reduced tensor $T_I(\cdot, \ldots, \chi^{(\ell)}) \in \mathbb{R}^{n_1 \times \cdots \times n_{\ell-1}}$ as a candidate tensor for the parent of $I$. Now let $J$ be a node of level $\ell - 1$. Its children have produced candidate tensors for $J$. Pick the candidate $T$ with the highest $|T(e_J)|$ to be $T_J$. In this way we have removed the lowest level of the tree and have assigned appropriate tensors to the new leaves.

Our goal is to prove that if we start with an $x$-large echelon tree, then with high probability the next echelon tree is $\delta x$-large. Inductively this would prove that with high probability over the choice of $\chi^{(1)}, \ldots, \chi^{(\ell)}$, we have $T_I(\chi^{(1)}, \ldots, \chi^{(\ell)}) \geq \delta^\ell$ for some leaf $I$ of the original echelon tree, completing the proof.

For a fixed node $J$ of level $\ell - 1$, we want to show that the quantity $T_I(e_J, \chi^{(\ell)})$ is at least $\delta x$ in magnitude for some child $I$ of $J$. But this is very similar to the $\ell = 1$ case of lemma 6, which we have already proved. The difference is that the pivots are not necessarily equal to 1, but are at least $x$ in magnitude. This implies that

$$\mathbb{P}[\forall I \text{ child of } J : |T_I(e_J, \chi^{(\ell)})| \leq \delta x] \leq p^{(1-c)n}.$$

The number of nodes at level $\ell - 1$ is at most $n^{\ell-1}$, so by a union bound, we get that with probability at least $1 - n^{\ell-1} p^{(1-c)n}$, the tree produced at the next level is $\delta x$-large (the union bound is over fewer than $n^{\ell-1}$ events, each corresponding to one $J$). Induction completes the proof. □

Now we give a proof of theorem 11. We use induction to prove a stronger version. Theorem 11 will be a corollary of the following by setting $\alpha_1 = \cdots = \alpha_\ell = 1 - c$.

**Theorem 12.** *If $W \subseteq \mathbb{R}^{n_1 \times \cdots \times n_\ell}$ is a subspace, and $\alpha_1, \ldots, \alpha_\ell \in [0,1]$ are such that*

$$(1 - \alpha_1)(1 - \alpha_2) \cdots (1 - \alpha_\ell) \geq 1 - \dim(W)/(n_1 \cdots n_\ell),$$

*then there is an echelon tree for $W$ with fractional branching $(\alpha_1, \ldots, \alpha_n)$ such that for each leaf $I$ we have $\|T_I\|_\infty = 1$ and $|T_I(e_I)| = 1$.*

*Proof.* We use induction on $\ell$. For the base case of $\ell = 1$, we have $\alpha_1 \geq \dim(W)/n_1$ and we want an echelon tree with branching factor $\alpha_1 n_1 \leq \dim(W)$. We have already proved this case.

Now assume we have proved the statement for $\ell - 1$ and want to prove it for $\ell$. Consider partially flattening the tensor space by merging the first two dimensions, i.e., considering $W$ as a subspace of $\mathbb{R}^{n_1 n_2 \times n_3 \times \cdots \times n_\ell}$. Let us fix $\beta \in [0, 1]$ such that the premise of the induction hypothesis holds and we can get an echelon tree of height $\ell - 1$ with fractional branching $(\beta, \alpha_3, \ldots, \alpha_\ell)$. Nodes at level 1 of this tree have indices in $[n_1 n_2]$, and there are $\beta n_1 n_2$ of them. Considering these indices as living in $[n_1] \times [n_2]$, by the pigeonhole principle at least $\beta n_1 n_2 / n_1 = \beta n_2$ of them will have the same first component; let's call this component $i_1 \in [n_1]$. We can now extract the subtrees of these $\beta n_2$ elements and join them into an echelon tree of height $\ell$. The common parent of these nodes will have index $i_1$. So far we have constructed an echelon tree of height $\ell$ with fractional branching $(1/n_1, \beta, \alpha_3, \ldots, \alpha_\ell)$.

Now consider the subspace $\{T \in W \mid T(e_{i_1}, \cdot, \ldots, \cdot) = 0\}$. We think of $W$ as living in $\mathbb{R}^{(n_1 - 1)n_2 \times n_3 \times \cdots \times n_\ell}$, since index $i_1$ has been eliminated from the first dimension. We can again apply the induction hypothesis to this space and as long as the premise holds obtain an echelon tree of height $\ell - 1$ with fractional branching $(\beta, \alpha_3, \ldots, \alpha_\ell)$. We can apply the pigeonhole principle again to find $\beta(n_1 - 1)n_2 / (n_1 - 1) = \beta n_2$ level-1 nodes having the same first index $i_2$. We extract a height $\ell$ echelon tree from them and join this with the height $\ell$ echelon tree we already have. At the end we will have an echelon tree with fractional branching $(2/n_1, \beta, \alpha_3, \ldots, \alpha_\ell)$.

Suppose we have repeated this procedure $\gamma n_1 - 1$ many times and currently have a height $\ell$ echelon tree with fractional branching $(\gamma, \beta, \alpha_3, \ldots, \alpha_\ell)$. As long as the premise of the induction hypothesis holds we can grow this echelon tree. The current subspace is $\{T \in W \mid W(e_{i_j}, \cdot, \ldots, \cdot) = 0 \text{ for } j \in [\gamma n_1]\}$ which lives in $\mathbb{R}^{(1-\gamma)n_1 n_2 \times n_3 \times \cdots \times n_\ell}$. The dimension of this subspace is at least $\dim(W) - \gamma n_1 n_2 \cdots n_\ell$. So the premise of the induction hypothesis holds as long as

$$(1 - \beta)(1 - \alpha_3) \cdots (1 - \alpha_\ell) \geq 1 - \frac{\dim(W) - \gamma n_1 \cdots n_\ell}{(1 - \gamma)n_1 n_2 \cdots n_\ell} = \frac{1 - \dim(W)/n_1 \cdots n_\ell}{1 - \gamma}.$$

This means that as long as $(1 - \gamma)(1 - \beta)(1 - \alpha_3) \cdots (1 - \alpha_\ell) \geq 1 - \dim(W)/n_1 \cdots n_\ell$, we can grow the echelon tree.

To finish the proof, we set $\beta = \alpha_2$, which means that while $\gamma < \alpha_1$, we can grow the echelon tree. So when this procedure stops we have an echelon tree with fractional branching $(\alpha_1, \ldots, \alpha_\ell)$.  $\square$

## 3.3  Implications for the main question

Our result, theorem 5, together with results from [3] (see the supplementary material), imply that under very mild assumptions we can recover $\mathcal{S}_1, \ldots, \mathcal{S}_n$ from their $\ell$-wise intersections as long as $|\mathcal{U}| \leq n^{\Theta(\ell)}$. These mild assumptions are necessary to prevent adversarially constructed examples that have no hope of unique recovery.

To get a sense of the mild assumptions that we need, let us discuss the parameters that appear in theorem 5. We assume that $\ell$ is a constant that does not grow with $n$. We can take $c$ to be some fixed constant as well. For example $1/2$, or even $1/\sqrt[\ell]{2}$. If we perturb our cell assemblies according to example 3, i.e., flip assembly memberships for each neuron class and assembly pair with probability $q$, how large of a $q$ do we need for the conditions of theorem 5 and [3] to be satisfied? The distribution we get for $\chi(u)$s is going to be $(1/2, 1 - q)$-nondeterministic as long as $q \leq 1/2$. So $\delta = 1/2$ is a constant. The only condition we need is now for the failure probability to be small. This roughly translates to

$$n^{O(\ell)}(1 - q)^{(1-c)n} \ll 1,$$

which will be satisfied for $q = \Omega(\log n / n)$. In other words, we only have to flip each coordinate of $\chi(u)$ with probability $O(\log n / n)$. On average, each neuron's membership will be changed in about $O(\log(n))$ of the assemblies, which is a very small fraction of the assemblies. For slightly larger values of $q$, e.g., $q = n^{\epsilon - 1}$, the probability of failure becomes exponentially small similar to [3].

We also assumed that $w(u) = 1$ for all $u \in \mathcal{U}$. In general this is not needed. As long as the weights $w(u)$ are in a range whose upper bound is at most a polynomially bounded factor larger than the lower bound, we can absorb the weights into the vector $\chi(u)$ and the running time and accuracy will only suffer by a polynomially bounded factor.

We also remark that recovering a $\{0, 1\}^n$ vector within an additive error of $1/n$ is the same as exact recovery (by rounding the coordinates). So by setting the recovery error (see supplementary material) to $1/n$ we get exact recovery.

Finally, we remark that even though we are mostly interested in the case where $\ell = O(1)$, our dependencies on $\ell$ seem to be better than the results of [3] even in the setting of Gaussian perturbations. In particular, our running time (as well as our tolerance for error) grows polynomially with $n^\ell$, whereas the running time of [3] grows with $n^{3^\ell}$. When adding Gaussian noise of total variance $\rho^2$ as in example 4, we can treat our vectors as coming from a $(O(\rho/\sqrt{n}), 1/2)$-nondeterministic distribution. This means our probability of failure will be at most $n^{2\ell}/2^{(1-c)n}$. To have a fair comparison, we need to allow for the number of components to be roughly half the total dimension, so we need to let $c = 1/\sqrt[\ell]{2} \simeq 1 - \Theta(1/\ell)$. So the probability of failure will be roughly $\exp(O(\ell \log n) - \Omega(n/\ell))$. For large enough values of $\ell$ this is much better than the guarantee of $\exp(-\Theta(n^{1/3^\ell}))$ of [3].

## 4 Association graphs and the soft model

When the number of observations is smaller than what is needed for reconstruction, we can still ask whether there exists *some* Venn diagram that is consistent with the observations. *Which classes of weighted graphs (or hypergraphs) can be represented by Venn diagrams?*

Interestingly, a similar model was formulated almost three decades ago, motivated by quantum mechanics and spin glass systems, and a mathematical object called *correlation polytope* was defined to frame that investigation [20]. It is not hard to show that membership in the polytope is an NP-hard problem and natural optimization variants of it are hard to approximate.

In this section we formulate a promise version of the problem where either the intersection is above a certain threshold (corresponding to association) or below another (corresponding to non-association) which seems to be more tractable.

More precisely, we are given a graph that is *unweighted*. The nodes still stand for assemblies of neurons, all of the same size $K$, out of a universe of $N$ neurons, and the edges signify association; the difference is that, in this model, if two assemblies are associated then they have an intersection of size at least $a$; whereas if they are not, then their intersection is at most $b$. The intended relationship between these numbers is that $N$ is much larger than $K$ (we take it to be a power of $K$), and $K$ is in turn much larger than $a$, while $a$ is quite a bit larger than $b$. To fix ideas, in the sequel we take $N = K^2$ and $b < a$ small constant fractions of $K$; in the experiment in [6, 15] $a$ and $b$ are found to be about 8% and 4% of $K$, respectively. We call a graph $G = (V, E)$ *representable* with parameters $(N, K, a, b)$ if every node of $G$ can be associated with a set of $K$ neurons such that for any two adjacent nodes the corresponding sets have intersection at least $a$, while for any two non-adjacent nodes the corresponding sets have intersection at most $b$. The question is, *which graphs are representable?*

**Theorem 13.** *Any graph of maximum degree at most $2K/a$ is representable, and so is any tree of maximum degree $2K^2/a^2$.*

The $2K/a$ bound follows from the fact that the edges of a regular Eulerian graph can be decomposed into cycles, while the $2K^2/a^2$ follows from the theory of block designs. Recalling that $a$ is a small fraction of $K$, we conclude that rather rich and complex "association graphs" can be represented *in principle*. But can these sophisticated combinatorial constructions be carried out with surgical precision in the wet chaos of the brain?

Here is a more realistic framework which we call *the soft model:* Suppose that we are given an association graph $G = (V, E)$. We wish to determine whether a model of $G$ exists, i.e., $|V|$ sets corresponding to nodes of $G$ whose pairwise intersections realize $G$ according to the rules above involving $a$ and $b$. We wish to create sets of expected size $K$ representing the nodes, starting from the universe of neurons $[N]$ and executing instructions of the following form (in the following $C, C_1, C_2$ are previously constructed sets, and $A$ is the set being constructed):

$$A \leftarrow C_1 \cup C_2, \qquad A \leftarrow C_1 \cap C_2, \qquad A \leftarrow C_1 - C_2, \qquad A \leftarrow S(C, p),$$

where by $S(C, p)$ we denote the result of sampling each node in set $C$ with probability $p$ — a simple and realistic enough primitive. The question is, which graphs can be realized in such a way that the intended relations between the nodes and their intersections are not corrupted, with high enough probability, by the randomness of the process? We can show the following:

**Theorem 14.** *Any graph with maximum degree $\frac{1}{e} \cdot \frac{K}{a}$ can be realized in the soft model with high probability.*

## Footnotes

[1] Or person, these are commonly known as "Jennifer Aniston neurons".

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
