[Supplementary Material]

# A   Reduction to linear independence

We now mention the main result of [3]:

**Theorem 15** ([3]). *Let $|\mathcal{U}| \leq n^{\lfloor \frac{\ell-1}{2} \rfloor}/2$ for some constant $\ell$. Assume that for each $u \in \mathcal{U}$ and $i \in [\ell]$, we choose a vector $\chi(u)^{(i)} \in \mathbb{R}^n$ by starting from an adversarially chosen vector $\chi(u)^{(i)}_*$ of norm at most 1 and adding a standard Gaussian noise with variance $\sigma^2/n$ to each coordinate of $\chi(u)^{(i)}_*$. Now define the order-$\ell$ tensor*

$$T := \sum_{u \in \mathcal{U}} \chi(u)^{(1)} \otimes \cdots \otimes \chi(u)^{(\ell)},$$

*and assume that we get as input $T + E$ where $E$ is an order-$\ell$ (measurement error) tensor, whose entries are bounded by $\epsilon(\sigma/n)^{3^\ell}$ for some $\epsilon < 1$. Then there is an algorithm that recovers all the tensors $\{\chi^{(1)}(u) \otimes \cdots \otimes \chi(u)^{(\ell)}\}_{u \in \mathcal{U}}$ up to an additive $\epsilon$ error. This algorithm runs in time $n^{O(3^\ell)}$ and succeeds with probability $1 - \exp(-O(n^{1/3^\ell}))$.*

The algorithm behind theorem 15 is based on a robust version of order-3 tensor decomposition, widely known as the "simultaneous diagonalization" or Chang's lemma [3, 5, 17]. Roughly speaking, the tensor

$$T = \sum_{u \in \mathcal{U}} \chi(u)^{(1)} \otimes \cdots \otimes \chi(u)^{(\ell)}$$

can be viewed as an order-3 tensor by grouping some of factors together:

$$T = \sum_{u \in \mathcal{U}} \underbrace{\left( \chi(u)^{(1)} \otimes \cdots \otimes \chi(u)^{((\ell-1)/2)} \right)}_{\text{factor}} \otimes \underbrace{\left( \chi(u)^{((\ell+1)/2)} \otimes \cdots \otimes \chi(u)^{(\ell-1)} \right)}_{\text{factor}} \otimes \underbrace{\chi(u)^{(\ell)}}_{\text{factor}}. \quad (2)$$

Then the algorithm from [3] depends on using a robust version of Chang's lemma to decompose $T$. It only needs the collection of first factors to be "robustly" linearly independent, the collection of the second factors to be "robustly" linearly independent, and the collection of the third factors to "robustly" not contain vectors parallel to each other (a weaker notion than linear independence). We give the precise required conditions below:

**Theorem 16** ([3]). *Consider the tensor*

$$T = \sum_{u \in \mathcal{U}} a(u) \otimes b(u) \otimes c(u)$$

*and assume that the following conditions are satisfied:*

1. *The condition numbers of the matrices $A, B$ are bounded by $\kappa$, where $A$ is formed by taking $a(u)$s as columns and $B$ by taking $b(u)$s as columns,*

2. *For any $u_1 \neq u_2$, the vectors $c(u_1)$ and $c(u_2)$ are far from being parallel: $\| \frac{c(u_1)}{\|c(u_1)\|} - \frac{c(u_2)}{\|c(u_2)\|} \| \geq \tau$,*

3. *All of the vectors $a(u), b(u), c(u)$ have norms bounded by $C$, a polynomially bounded quantity.*

*Then there is an efficient algorithm, running in time $\mathrm{poly}(\kappa, 1/\tau, n^\ell)$, that recovers $a(u) \otimes b(u) \otimes c(u)$ for all $u$ within additive error $\epsilon$, by only observing $T + E$ where $E$ is a noise tensor whose entries are bounded by $\epsilon \cdot \mathrm{poly}(1/\kappa, 1/n^\ell, \tau)$.*

Condition 1 is arguably the most difficult one to satisfy. Condition 2 is satisfied with high probability for many distributions of interest $\mathcal{D}$, but it can also be automatically reduced to condition 1 if one is willing to change the grouping in eq. (2). If instead of having three groups, the first two composed of $(\ell-1)/2$ factors and the last one composed of one factor, we create three equal-sized groups (each consisting of $\ell/3$ factors), then the last group would also have a bounded condition number (by an extension of condition 1) and will automatically satisfy condition 2. This makes the dependency on $\ell$ worse but would still give us something similar to theorem 15 with $|\mathcal{U}| \leq n^{\lfloor \frac{\ell-1}{2} \rfloor}/2$ replaced by

$|\mathcal{U}| \leq n^{\lfloor \frac{\ell}{3} \rfloor}/2$. Finally, note that condition 3 is also automatically satisfied with very high probability for Gaussian perturbations and also our model, in which we sample vectors from the hypercube $\{0, 1\}^n$. We assume that $\mathcal{D}$ is not only $(\delta, p)$-nondeterministic but also that it satisfies condition 3 with high probability.

In section 3 we focus only on proving condition 1 in theorem 16. To make the notation simpler we replace $\ell$ by $(\ell - 1)/2$, and assume $a(u)$s are tensors of $\ell$ factors. In order to bound the condition number of $A$ in theorem 16, we need to lowerbound the minimum singular value and upperbound the maximum singular value of $A$. An upperbound on $\sigma_{\max}(A)$ is readily given by condition 3 of theorem 16. The matrix $A$ has columns with norms bounded by $C$ and therefore

$$\sigma_{\max}(A) \leq n^{\ell/2}C.$$