[Reviews · NeurIPS 2018]

Reviewer 1



This paper that is built on top of Bhaskara etal. [3 in the main paper] described an analysis linear independence of rank one tensors produced by tensors powers of randomly perturbed vectors and allowed for discrete perturbations. Moreover, the described analysis is related to the recovery of assemblies of neurons. Overall, the paper gave a detailed analysis of the tensor decomposition using l=1and a general case although a better differentiation with respect to [3] is needed. Moreover, the proposed application of recovering assemblies of neurons is misleading. For instance, the application aim is not clear enough because the abstract mentioned recovering assemblies of neurons but on the main text and problem formulation aims to talks about assembly association and the structure intersection of cell assembly intersection, and also how the problem is posed since there is ambiguous definitions, e.g. what do you mean about m classes of neurons? Or entities or concepts? Moreover, what would it happen if there is no association among neurons but there is still an overlap as have been shown in different papers [1,2]? And how it affects the analysis if we do not have the same number of neurons for each ensemble? Therefore, a detailed experiment of applying the proposed approach would have a great impact into the neuroscience community. Minors comments: Footnote 1. Replace … for . Line 163. Typo Gassuian. References: [1] Luis Carrillo-Reid, Jae-eun Kang Miller, Jordan P. Hamm, Jesse Jackson and Rafael Yuste. Endogenous Sequential Cortical Activity Evoked by Visual Stimuli. Journal of Neuroscience 10 June 2015, 35 (23) 8813-8828; [2] Eleonora Russo , Daniel Durstewitz. Cell assemblies at multiple time scales with arbitrary lag constellations. Elife 2017. =================================================== I appreciate the author effort of addressing my concerns, but after reading carefully the author’s feedback I keep my score because there is a weak connection to application of cell assemblies association or not very clearly written as mentioned by R2 “it could spend more time explaining the relationships between the assemblies problem, the venn diagram formalization and tensor decomposition”. Therefore, I was interested how could the proposed approach deal no concept association among neurons besides having an overlap as have been shown in different papers [1,2]? and how it affects the analysis if we do not have the same number of neurons for each ensemble? and how will a neuroscientist be benefit using such approach without a direct formulation or analysis on real data?

Reviewer 2



This paper gives a new smoothed analysis result on tensor powers of randomly perturbed vectors. This may sound very technical, but as the paper mentioned tensor decompositions have been widely applied to learning many latent variable models, and in many cases they provide the only algorithms with theoretical guarantees. To apply tensor decomposition algorithm, it is common to require that tensor powers of the components are linearly independent (and in fact even the smallest singular value is large). This paper generalizes result of Bhaskara et al. and showed that for fairly general types of perturbations (which they summarized using (\delta, p) nondeterministic property), the perturbed rank-1 tensors will be (robustly) linearly independent. This could lead to better sample complexity guarantees for learning many latent variable models. The particular motivation in this paper comes from a problem of association of assemblies. In this application the components of the tensors need to come from a discrete set (the hyper-cube), which rules out the previous result by Bhaskara et al. The techniques in this paper gives a satisfying solution. At a high level the proof technique of this paper is similar to Bhaskara et al. as they look at the projection of each tensor into the orthogonal subspace of existing tensors, and try to construct certificates that show the projection is large. However at a more detailed level the proof is novel. Both the (\delta, p) nondeterministic property and the echelon tree idea seem novel and applicable in other settings. The first part of this paper is not very clearly written as it could spend more time explaining the relationships between the assemblies problem, the venn diagram formalization and tensor decomposition. The proof on the other hand is very clear. ==================================== After the response: I will keep my score but as reviewer 1 pointed out the application on cell assembly is confusing and probably not very convincing. I have no expertise in the application so my review is purely based on the tensor decomposition part. If this paper were to be accepted I hope the authors will make an effort in improving the presentation.

Reviewer 3



# Review for "Smoothed Analysis of Discrete Tensor Decomposition and Assemblies of Neurons" This paper models and studies the following problem. The brain is known to store "concepts" (the Eiffel tower, Barak Obama) as collections of neurons which fire simultaneously when the concept is "sensed" -- e.g. when one sees a picture of Barak Obama. Furthermore, if there are associations among concepts -- e.g. one has seen a picture of Barak Obama standing in front of the Eiffel tower -- then the "Barak Obama" neurons will tend to fire with the Eiffel tower ones. The high-level question is: is it possible to efficiently learn a "concept map" of someone's brain by measuring strength of associations among small sets of concepts? The authors model this question as one of learning a set system from (i.e. the Venn diagram of a collection of sets S_1...S_n over some universe) from the sizes of all $\ell$-wise intersections, for some "reasonable" $\ell$. The main result is a polynomial-time algorithm accomplishing this learning problem, so long as 1. set memberships are slightly randomly perturbed (a la smoothed analysis) and 2. the number of measurements $n^\ell$ is larger than the size of the underlying universe. The algorithm assembles these $n^\ell$ measurements into a certain $\ell$-tensor, then applies known tensor decomposition algorithms to recover the set system. The catch is that these tensor decomposition results are only known to apply to in a smoothed-analysis setting with *Gaussian* perturbations to the underlying tensor components. This is not a natural kind of perturbation in this discrete, set-system setting. The authors show that these tensor decomposition algorithms also work under a much wider variety of perturbation models, essentially any random perturbation for which the each coordinate's perturbation remains slightly anticoncentrated even conditioned on the perturbations of all other coordinates. This technical result on allowable perturabtion models for smoothed analysis is of interest in its own right -- smoothed analyses for high-dimensional geometric problems are few and far between because they pose significant technical challenges, and the authors seem to have invented some interesting new tools to carry out their analysis. I am not an expert in this kind of neural modeling, so I do not know whether similar models and algorithmic questions have been addressed in prior literature. I am an expert in tensor decomposition algorithms; in my view the new smoothed analysis represents a NIPS-worth contribution in itself, especially since the application is reasonably interesting. Tensor decomposition is a major tool in provable algorithms for machine learning: expanding the class of smoothed analyses under which tensor decomposition methods are known to work removes unnecessary and unnatural technical assumptions on the kinds of models which can be learned by tensor decomposition. The writing is clear in the first few pages; I did not have the chance to verify the technical details line-by-line but the approach described seems reasonable. I recommend in favor of accepting this paper to NIPS.